

# First report of the ectomycorrhizal fungal community associated with two herbaceous plants in Inner Mongolia, China

Yongjun Fan[1,2], Simin Xiang[3], Jing Wang[4], Xuan Zhang[3], Zhimin Yu[1], Shupeng Zhu[3], Meng Lv[3], Lijun Bai[3], Luyu Han[3], Jianjun Ma[5] and Yonglong Wang[3]

[1] School of Life Science and Technology, Inner Mongolia University of Science and Technology, Baotou, Inner Mongolia, China
[2] Yinshanbeilu Grassland Eco-hydrology National Observation and Research Station, China Institute of Water Resources and Hydropower Research, Inner Mongolia, China
[3] Faculty of Biological Science and technology, Baotou Teacher's College, Baotou, Inner Mongolia, China
[4] Department of Civil Engineering, Ordos Institute Technology, Ordos, Inner Mongolia, China
[5] College of Life Science, Lang Fang Normal University, Lang Fang, Hebei, China

Corresponding authors
Jianjun Ma,
maandyao8184@163.com
Yonglong Wang,
wylongceltics@163.com

## ABSTRACT

Ectomycorrhizal (EM) fungi play a vital role in ensuring plant health, plant diversity, and ecosystem function. However, the study on fungal diversity and community assembly of EM fungi associated with herbaceous plants remains poorly understood. Thus, in our study, *Carex pediformis* and *Polygonum divaricatum* in the subalpine meadow of central Inner Mongolia, China were selected for exploring EM fungal diversity and community assembly mechanisms by using llumina MiSeq sequencing of the fungal internal transcribed spacer 2 region (ITS2). We evaluated the impact of soil, climatic, and spatial variables on EM fungal diversity and community turnover. Deterministic *vs.* stochastic processes for EM fungal community assembly were quantified using $\beta$-Nearest taxon index scores. The results showed that a total of 70 EM fungal OTUs belonging to 21 lineages were identified, of which *Tomentella-Thelephora*, Helotiales1, *Tricholoma, Inocybe, Wilcoxina* were the most dominant EM fungal lineages. EM fungal communities were significantly different between the two herbaceous plants and among the two sampling sites, and this difference was mainly influenced by soil organic matter (OM) content and mean annual precipitation (MAP). The neutral community model (NCM) explained 45.7% of the variations in EM fungi community assembly. A total of 99.27% of the $\beta$-Nearest Taxa Index ($\beta$NTI) value was between −2 and 2. These results suggest that the dominant role of stochastic processes in shaping EM fungal community assembly. In addition, $RC_{bray}$ values showed that ecological drift in stochastic processes dominantly determined community assembly of EM fungi. Overall, our study shed light on the EM fungal diversity and community assembly associated with herbaceous plants in the subalpine region of central Inner Mongolia for the first time, which provided a better understanding of the role of herbaceous EM fungi.

## INTRODUCTION

Ectomycorrhizal (EM) fungi are crucial for the growth and survival of a number of ecologically and economically important plant species in boreal, temperate, and arctic regions (*Smith, 2009*; *Anthony et al., 2022*; *Averill et al., 2022*; *Cahanovitc et al., 2022*). In these environments, EM fungi can not only promote the nutrient absorption and stress resistance of host plants, but also improve the soil environment, stabilize and improve the soil fertility level, and rely on the soil environment to obtain the nutrients needed for growth through host plants (*Tedersoo et al., 2008*; *Smith, 2009*; *He, Cui & Ma, 2021*; *Averill et al., 2022*). Therefore, the research on the community composition and maintenance mechanism of EM fungi can better understand the significance of biological community construction and ecosystem function, and can also make the EM fungi resources truly expand and utilize.

How are biological communities constructed and which species can coexist in the same habitat, which has been the core issue of ecological research (*Vályi et al., 2016*). There are two classical theories to explain the process of community construction: niche theory and neutral theory. Community neutrality theory emphasizes the influence of random process on community construction, and niche theory emphasizes the role of deterministic process. Deterministic processes derive from environmental filtration or environmental heterogeneity (*Wang et al., 2013*). Although the two theories are obviously different, they have successfully explained the construction of biological communities. In recent years, more and more studies have confirmed that the biological community under natural conditions is determined by deterministic niche related processes and stochastic neutral processes (*Wang et al., 2019*; *Liu et al., 2020*; *Wang et al., 2021*). The deterministic process is mainly determined by abiotic factors (*i.e.*, climatic conditions and soil factors) and biotic factors (host plant specificity), which represents the influence of environmental choice on the composition of EM fungal community. It has been found that abiotic factors (soil factors, precipitation) are the decisive factors for EM fungal community compositions (*Mühlmann, Bacher & Peintner, 2008*; *Blaalid et al., 2012*; *Davey et al., 2015*; *Mundra et al., 2015*; *Koizumi & Nara, 2020*; *Sugiyama, Matsuoka & Osono, 2021*). The research on the EM fungi associated with *Kobresia capillifolia*, *Carex parva*, *Polygonum macrophyllum*, and *Potentilla fallens* distributed in the Daxueshan, Hongshan, and Baima Snow mountains showed that EM fungi have a preference for host plants (*Gao & Yang, 2016*). The stochastic process is dispersal limitation, ecological drifts, genetic mutation, and historical contingencies (*Dini-Andreote et al., 2015*; *Liu et al., 2020*). Previous studies have shown that dispersal limitation can affect the community composition of EM fungi on different spatial scales, including local scale (*Peay, Garbelotto & Bruns, 2010*; *Gao et al., 2014*, *2017*), regional scale (*Tedersoo et al., 2011*; *Bahram et al., 2012*; *Van der Linde et al., 2018*; *Wu et al., 2018*), intercontinental scale (*Glassman, Wang & Bruns, 2017*) and global scale (*Põlme et al., 2013*), However, there are differences between different spatial scales. such as the researches on the EM fungi related to Betulaceae in China's secondary forest ecosystem and five natural pine species distributed in Inner Mongolia area showed that the community is determined by the dispersal limitation (*Wang et al., 2019*, *2021*). In recent

years, many scholars have found that EM fungi with different colonization degrees exist in the roots of some herbs such as genus of *Carex*, *Polygonum*, *Kobresia* and *Potentilla* (*Mitchell & Harrington, 2002*; *Wang & Qiu, 2006*; *Li & Guan, 2007*; *Gao & Yang, 2016*). *Yao et al. (2013)* analyzed the diversity and community composition of EM fungi in the root system of *P. viviparum* in the mountains of central Norway, and found that the diversity of EM fungi in both snow and mountain ridges was positively correlated with soil phosphorus content. *Thoen et al. (2019)* also found that the EM fungal community from different parts of the root system of *P. viviparum* had obvious spatial structure on a single root system, and the EM fungal community moved from the youngest end of the rhizomes to the older end. Thus, it should be noticed that the studies focused on EM fungal communities associated woody plants received much more attention than herbs, particularly with regard of mechanisms underlying community assembly. *Carex pediformis* and *Polygonum divaricatum*, as the dominant herbs in the subalpine area of Inner Mongolia, China, play an important role in maintaining the subalpine ecosystem and biodiversity. However, the distribution pattern and diversity of EM fungi in the root system of the dominant herbs in the subalpine meadow area in the middle of arid and semi-arid Inner Mongolia remains unclear now.

Understanding the diversity and distribution pattern of EM fungi in this area is of great significance to plant protection and ecosystem construction. In recent years, with the further development of molecular biology, the amplicon sequence of EM fungi was obtained by high-throughput sequencing technique. By analyzing the diversity and community assembly of EM fungi, as well as the correlation between EM fungi and host plants, soil, climate, spatial distance, *etc*., it is helpful to reveal the geographical distribution pattern and maintenance mechanism of EM fungi in herbaceous roots. In this study, we have investigated the diversity and community assembly of EM fungi in the roots of *C. pediformis* and *P. divaricatum* in central Inner Mongolia, China. We hypothesize that (i) both the EM fungal diversity and community composition are significantly different between the two herb species, and the two sites; (ii) stochastic processes play more important roles than deterministic process in driving EM fungal community assembly. Thus, the aims of our study are to reveal: (i) the characteristics of the diversity of EM fungi associated with *C. pediformis* and *P. divaricatum*; (ii) the differences in the community assembly of EM fungi associated with *C. pediformis* and *P. divaricatum* and influence factors; (iii) and assess the importance of ecological processes for EM fungi associated with herbaceous plants.

## MATERIALS AND METHODS

### Research filed and sample collection

#### Study area

*C. pediformis* and *P. divaricatum* were selected from the subalpine meadow areas of Chunkun Mountain (CKS) and Jiufeng Mountain (JFS) of central Inner Mongolia of China in this study. The sampling areas (CKS and JFS) is located in the middle of Yinshan Mountain range. The mountain runs east-west and becomes the natural boundary between

the Loess Plateau and the Mongolian Plateau. Its plant division belongs to the North China broad-leaved forest mountain area in the Eurasian grassland area (*Comprehensive Survey Team of Inner Mongolia & Ning Xia of the Chinese Academy of Sciences, 1985*). The average altitude is 1,500–2,000 m and Xijiufeng is the peak (2,338 m). Alpine meadows are distributed in the alpine belt more than 2,000 m above of Jiufeng Mountain and Chunkun Mountain. The alpine meadow does not form a belt in the vertical band spectrum, but appears in patches on the top of the mountain about 2,000 m, forming a hat shaped alpine meadow belt. According to the extracted climate information from the WorldClim dataset at 30-arc-second resolution (*Hijmans et al., 2005*), the mean annual temperature (MAT) range is 0.21–4.58 °C, and the mean annual precipitation (MAP) range is 313–360 mm. Use a highly sensitive GPS instrument (M-241; Holux Technology Inc., Taiwan, China) to record geographic coordinates (Latitude and Longitude). Information on the geographic locations and climate is summarized in Table S1.

*Sample collection*

Our field work was performed on September 12, 2020, and October 18, 2020; at each site, four to seven individuals in each location were selected. The individuals should be at least 10 m apart to ensure sample independence (*Pickles et al., 2012*). The whole plants were excavated using a shovel, and then the roots were cut using shovels. Meanwhile, at each location, samples of rhizosphere soil were collected from each individual and then combined into a composite sample. Root samples were labeled and placed in the autoclaved article envelopes, then transported to the laboratory in an ice box and stored in a −20 °C refrigerator. The soil samples were air-dried and passed through a 2 mm sieve for analysis of chemical properties. In this study, 23 root samples and 10 soil samples were obtained.

## Analysis of soil properties

Soil chemical properties include pH, available phosphorus (AP), available nitrogen (AN), available potassium (AK), organic matter (OM), total carbon (TC), total phosphorus (TP), total potassium (TK) and total nitrogen (TN) were measured in this study. After mixing dry soil and distilled water in a ratio of 1:2.5 (w/v), the pH of the soil was measured with a five easy pH meter. A soil sample nutrient rapid measuring instrument (TPY-8A; China Top Yunnong Co., Ltd., Shanghai, China) was used to measure soil AP, AN, AK and OM. TC and TN were measured by direct combustion using a Vario EL III C/N Element Analyzer (Elementar Analysensysteme GmbH, Langenselbold, Germany). TP and TK were measured using an iCAP 6300 inductively coupled plasma spectrometer (Thermo Scientific, Waltham, MA, USA). The analysis of TC, TN, TP and TK were conducted by China Jisi Huiyuan Company. The information on soil properties is summarized in Table S1.

## DNA extraction, PCR, and MiSeq sequencing

Before molecular analysis, the root samples were carefully washed with tap water. Under a stereo microscope to pick EM root tips. In total, 100 healthy root tips were randomly selected from each sample. As a result, 2,300 EM root tips were sampled from 23 samples

for DNA extraction. An improved modified cetyltrimethylammonium bromide (CTAB) method was used to extract total genomic DNA (*Gao et al., 2014*). A two-step polymerase chain reaction (PCR) procedure has been adopted to amplify the fungal ITS2 region in the Veriti 96-well Thermal Cycler (Applied Biosystems, Waltham, MA, USA). Specific experimental steps are the same as described in *Wang et al. (2021)*.

## Bioinformatics analysis

The raw data used QIIME Pipeline-Version 1.7.0 (*Caporaso et al., 2010*) for quality control. Eliminate the sequence of >6 ambiguous bases and invalid primers; secondly, ITSx software (*Bengtsson-Palme et al., 2013*) was used to extract the ITS2 region of the fungi. The Usearch v.11 was used to check chimera, and command to remove the potential chimera sequence (*Peay, Garbelotto & Bruns, 2010*). Based on the high-quality non-chimera ITS2 sequence obtained from the above process, the Usearch v.11 was used to perform the operational taxonomic units (OTU) assignment according to the sequence similarity of 97%. The representative sequence of each OTU (the largest number of sequences) is compared with BLAST (basic local alignment search tool) using INSDC (international nucleotide sequence databases collaboration) and UNITE as reference databases (*Altschul et al., 1990*). Use the identification standard of *Tedersoo et al. (2014)* as a reference for the identification of fungal OTU, that is, according to the similarity value greater than 90%, 85%, 80%, and 75%, the fungal OTU can be identified to the classification level of genus, family, order, and class. According to Tedersoo's nomenclature method (*Tedersoo & Smith, 2013*), EM fungi are identified if the OTUs were highly similar to known EM fungi sequences and the lineages were also identified. In order to avoid the influence of the difference in the number of sequences between samples on the subsequent analysis results, the EM fungal data using the subsample command the rarefy function in the ape package (*Paradis, Claude & Strimmer, 2004*) in R v3.6.1. The original sequences of 23 samples have been uploaded to SRA database of NCBI (Accession ID: SAMN26982404–SAMN26982426). Information on EM fungi in the present study is shown in Table S2.

## Statistical analysis

The EM fungal OTU accumulation curves for two herbs were plotted by using the specaccum command in the vegan package. Use the bartlett.test function in the Stats package (*R Core Team, 2019*) to test the homogeneity of the data. The alpha diversity indices were calculated for each sample in the vegan package (*Oksanen et al., 2013*), which included OTU richness, Chao1, and Shannon index. The non-parametric Kruskal-Wallis test were used to compare the difference of alpha diversity between two herbs, and then the wilcox test were used to conduct multiple comparisons between each plant species using the compare means function in the PMCMR package (*Wang et al., 2019*; *Zhang et al., 2022*). Multiple model reference method was adopted to investigate the significant factors for EM fungal diversity by using the GLM MuMIn package (*Bartón, 2018*). Principal coordinate analysis (PCoA) was performed to test for differences in EM fungal communities in different locations or plants based on Bray-Curtis distance using

Hellinger-transformed abundances of EM fungal OTUs. The significance was determined by permutational multivariate variance analysis (PerMANOVA) of different sites or plants by adonis command. Subsequently, the envfit function in vegan package was used to fit geographical location, climatic conditions and soil variables in non-metric multi-dimensional scaling (NMDS) analysis to determine the main factors affecting EM fungal communities, and PERMANOVA was used to test the significance of environmental factors to EM fungi. The corrt.test function of the psych package (*Revelle, 2021*) was used to calculate the correlation between the EM fungal species lineages and environmental factors.

The Neutral Community Model (NCM) was used to evaluate the potential importance of stochastic processes for EM fungal community assembly according to *Sloan et al. (2006)*, in which $R^2$ and m value indicate the goodness of fit of the model and the estimated migration rate, respectively. The $\beta$NTI and RC-bray values for EM fungal community were calculated to further quantify the importance of stochastic and deterministic processes in determining EM fungal community assembly. Briefly, firstly, the $\beta$-mean-nearest taxon distance ($\beta$MNTD) was calculated using the picant package (*Kembel et al., 2010*) and then implemented to calculate the $\beta$NTI based on null distribution of $\beta$MNTD. Further, deterministic and stochastic processes were partitioned into five ecological processes based on both $\beta$NTI and Bray–Curtis-based Raup-Crick Index ($RC_{bray}$) values, and the criteria to examine the relative roles of ecology processes was described previously in *Wang et al. (2021)*. All statistical analyses were performed in R v. 3.6.3 (*R Core Team, 2019*).

## RESULTS

### Ectomycorrhizal fungal database summary

After quality filtering, 1,475,980 non-chimera ITS2 sequences were obtained from 1,551,935 raw sequences. These high-quality sequences clustered into 1,336 non-singleton OTUs, of which 101 OTUs (84,938) were identified as EM fungi. After the rarefication, a normalized dataset contained 70 OTUs were still retained for subsequent analysis (Table S3). Among the 70 OTUs, 16 OTUs belonged to Ascomycota (35.85% of total EM fungal reads), 54 OTUs to Basidiomycota (64.15%).

### Ectomycorrhizal fungal diversity

The accumulation curves of each plant species did not show any signs of reaching an asymptote, suggesting that further sample collection may result in more unknown EM fungal OTUs (Fig. 1A). The abundance and frequency ranking of EM fungal OTU shows that the EM fungal community contains a small number of dominant lineage and a large number of rare lineage (Fig. S1). Richness, Shannon, and Chao1 index showed that there are significant differences in the alpha diversity of EM fungi in different herbs ($P < 0.01$, Figs. 1B–1D). In addition, there was no significant difference in EM fungi of the same herb in different mountain areas ($P > 0.05$, Figs. 1B–1D). GLM analysis indicated that MAP and altitude were significant factors for EM fungal diversity (Table 1).

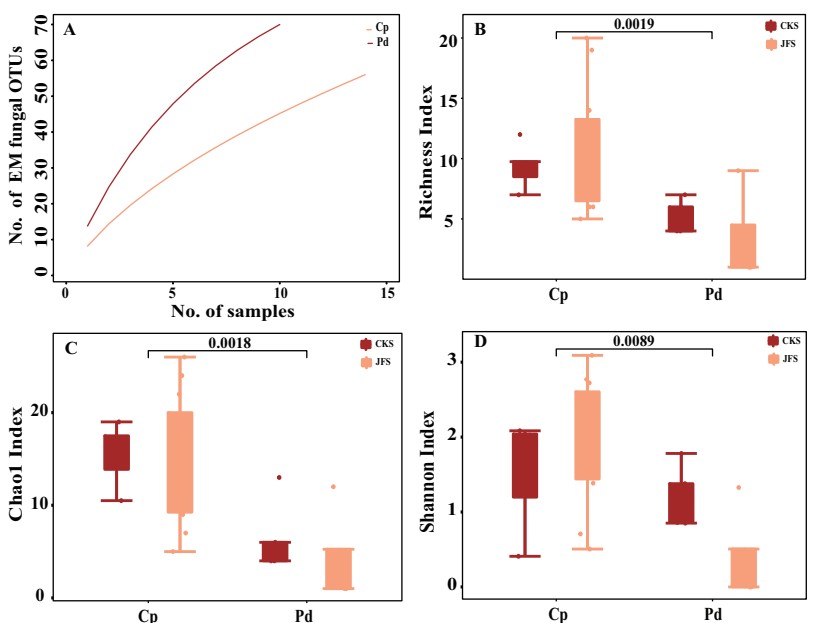

**Figure 1 Ectomycorrhizal (EM) fungal diversity of two herbs.** Accumulation curves of EM fungal operational taxonomic units (OTUs) (A), EM fungal Richness index (B), Chao1 index (C) and Shannon index (D) of two herbs. According to Wilcox test, the data on the column indicates significant differences among species ($P < 0.01$). Note: Pd, *Polygonum divaricatum*; Cp, *Carex pediformis*; JFS, Jiufeng Mountain; CKS, Chunkun Mountain.

**Table 1 GLM analysis indicate significant factors for EM fungal diversiry.**

|  | Estimate | Std. Error | Adjusted SE | z value | Pr(>\|z\|) |
|---|---|---|---|---|---|
| (Intercept) | 14.853 | 20.853 | 21.096 | 0.704 | 0.481 |
| MAP | −0.155 | 0.064 | 0.068 | 2.275 | 0.023* |
| Altitude | −0.010 | 0.004 | 0.005 | 2.111 | 0.035* |

**Note:**
Significance: *$P < 0.05$.

## Ectomycorrhizal fungal community composition and assembly

The total of 21 EM fungal lineages were found in the root samples of two herbs, of which *Tomentella-Thelephora* (47.58%), Helotiales1 (32.43%), *Tricholoma* (8.18%), *Inocybe* (5.15%), *Wilcoxina* (2.19%), (Table S4, >1% of total sequences) are the most dominant EM fungal lineages. *Serendipita*1, *Suillus-Rhizopogon*, *Pustularia*, *Cenococcum*, *Laccaria*, *Cantharellus*, *Pachyphloeus-Amylascus*, *Cortinarius*, *Meliniomyces*, *Boletus*, *Hebeloma-Alnicola*, *Hygrophorus*, *Amphinema-Tylospora* were only detected of *C. pediformis* (Fig. 2); *Tomentella-Thelephora*, Helotiales1, *Inocybe*, *Tricholoma* were only detected of *P. divaricatum*. The correlation analysis between the lineages of the EM fungi and environmental factors showed that K and MAP are positively correlated with *Tomentella-Thelephora*, MAP is negatively correlated with *Inocybe*, N, Latitude and TP are positively correlated, and Longitude and TC are negatively correlated with *Pachypleus-Amylascus* ($P < 0.05$; Fig. 3).

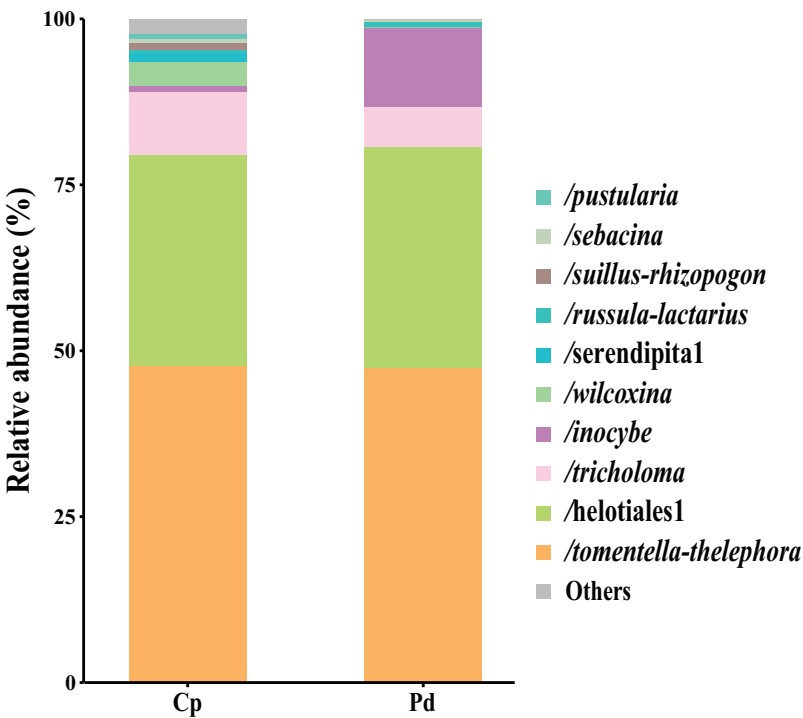

**Figure 2 EM fungal lineages and their relative abundance.** Here only showed abundant lineages (top 10 of all sequences). Note: Cp, *Carex pediformis*; Pd, Polygonum divaricatum.

The EM fungal community structure with the different sampling sites and herbs was analyzed using PCoA based on the Bray-Curtis distance (Fig. 4). PCoA ordination analysis showed that the PCoA first-axis explanation rate was 35.47%, the second-axis explanation rate was 19.76%, in which clear area separation in terms of herbs and locations. PerMANOVA agreed with the PCoA, in that, significant differences in the EM fungal communities between different herbs (Fig. 4, Adonis: $R^2$ = 0.268, $P$ = 0.01) and between two sampling sites (Fig. 4, Adonis: $R^2$ = 0.101, $P$ = 0.001). Based on the NMDS ordination analysis, the envfit test analysis showed that the OM (Adonis: $R^2$ = 0.301, $P$ < 0.05) and MAP (Adonis: $R^2$ = 0.28, $P$ < 0.05) are the most important factors affecting the EM fungal Community composition among two sites (Fig. 5, stress = 0.069).

The ecological assembly processes of EM fungal community were analyzed based on the NCM and $\beta$NTI. Whereas, the results of NCM analysis indicated that the assembly process of EM fungal communities was mainly affected by stochastic processes ($R^2$ = 0.457, m = 0.006; Fig. S2). In addition, most $\beta$NTI values fall in the range of −2 to 2 (99.27%; Fig. 6), while the case of |$\beta$NTI|>2 is only 0.73%, which also indicates that the assembly process of EM fungal communities is mainly determined by stochastic processes. Meanwhile, The result of calculating the distribution of |RC$_{bary}$| shows ecological drift, homogenizing dispersal and homogeneous selection account for 89.1%, 10.18% and 0.73% respectively (Fig. 7), indicating that stochastic processes had a stronger effect on the community assembly of EM fungi than the deterministic processes in the present study.

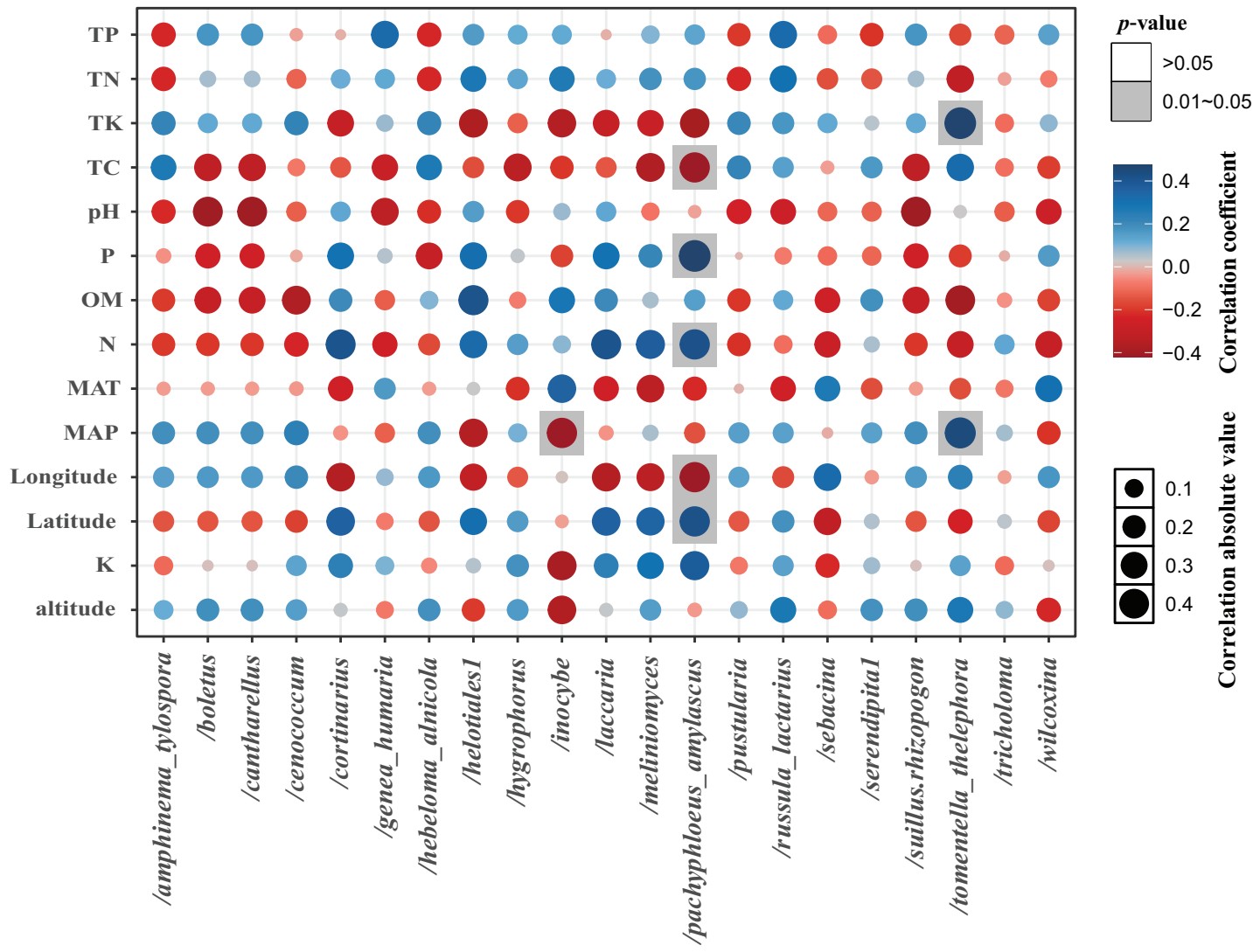

**Figure 3 The correlation between each lineage and environmental factors.** Including pH, available phosphorus (AP), available nitrogen (AN), available potassium (AK), organic matter (OM) content, total carbon (TC), total phosphorus (TP), total potassium (TK) total nitrogen (TN), mean annual temperature (MAT), mean annual precipitation (MAP), Longitude, Latitude, and altitude.

# DISCUSSION

The total of 70 EM fungal OTUs were identified from 23 samples in present study. And the richness index and the shannon index were 7.96 ± 1.72 (mean ± S.E.) and 1.4 ± 0.74, respectively. The species accumulation curve did not reach the plateau, implying our sample size was not adequate. Thus, more samples should be collected in our future study, which may introduce higher fungal diversity.

The analysis of the species composition of the EM fungi in this study showed that Serendipita1, *Suillus-Rhizopogon*, *Pustularia*, *Cenococcum*, *Laccaria*, *Cantharellus*, *Pachyphloeus-Amylascus*, *Cortinarius*, *Meliniomyces*, *Boletus*, *Hebeloma-Alnicola*, *Hygrophorus*, *Amphinema-Tylospora* are unique to *C. pediformis*. These results indicate that the host plant has a significant impact on the species composition of EM fungi (*Gao &*

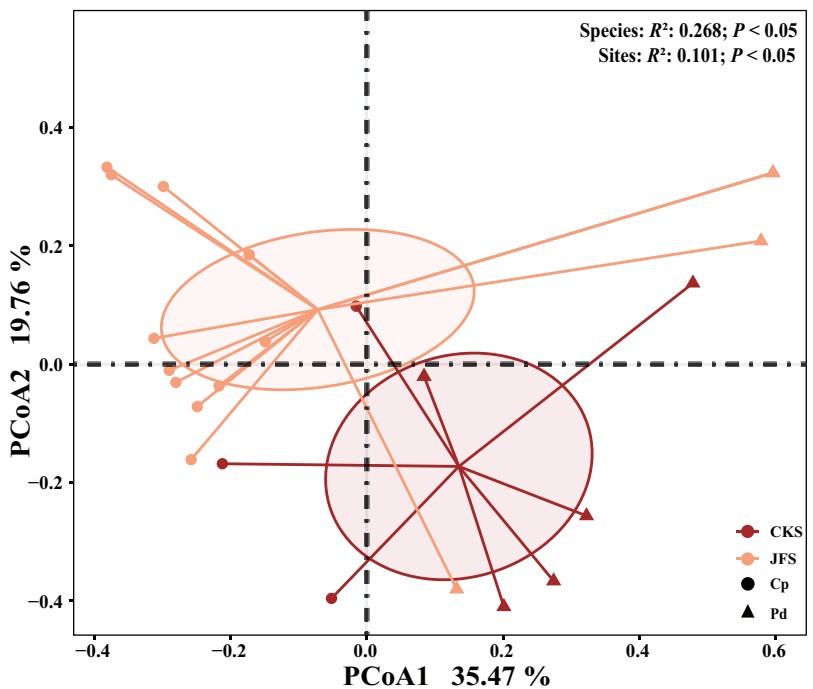

**Figure 4 EM fungal community beta-diversity visualized using PCoA ordination based on the Bray-Curtis similarity.** EM fungal were clustered and the center of gravity computed for each site. JFS, Jiufeng Mountain; CKS, Chunkun Mountain.

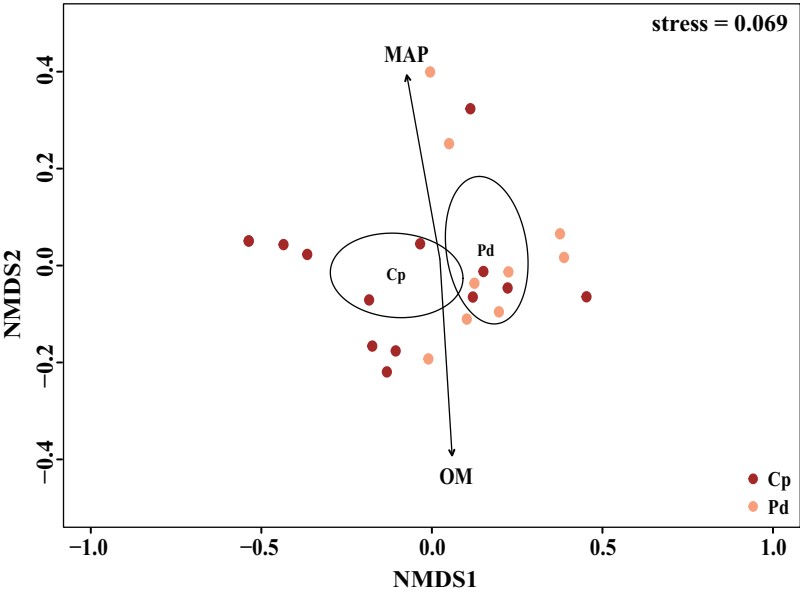

**Figure 5 Non-metric multidimensional scaling (NMDS) of the EM fungal community composition (Sorensen distance).** The ellipse represents the 95% confidence interval around the center point of each sites. OM, soil organic matter; MAP, mean annual precipitation; JFS, Jiufeng Mountain; CKS, Chunkun Mountain.
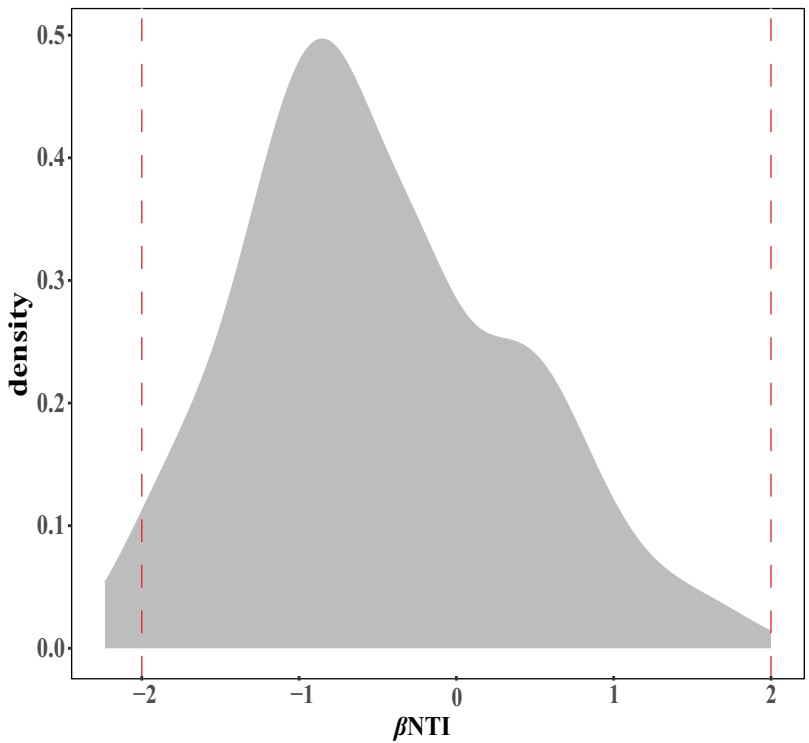

**Figure 6 EM fungal ecological processes.** $\beta$-nearest taxon index ($\beta$NTI) of EM fungal OTUs.

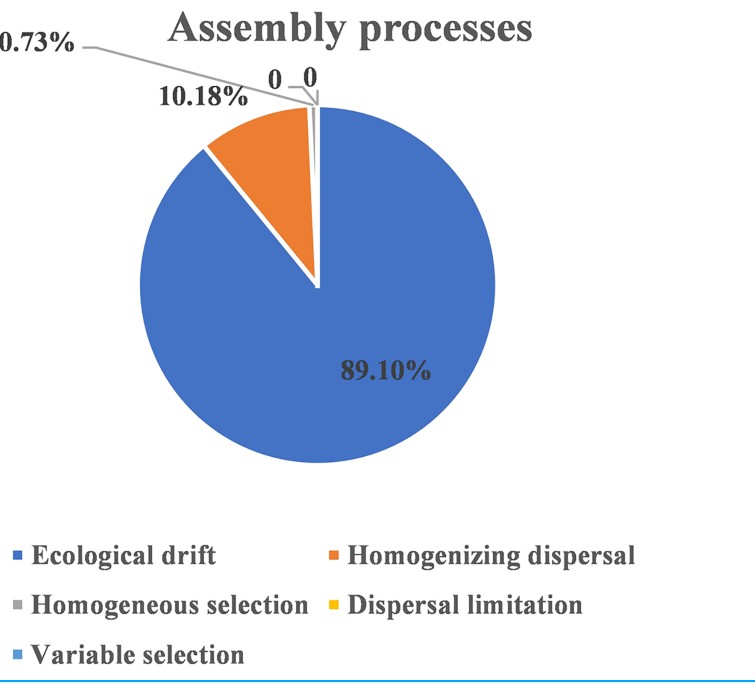

**Figure 7 EM fungal ecological processes.** Pie chart of community construction process calculated by
$RC_{bary.}$.

*Yang, 2016*). One explanation is the conservatism of the host phylogenetic niche (*Losos, 2008*). In detail, closely related plant species commonly showed more similarity than distantly related species in morphological and functional traits and appeared to share more similar fungal groups. *Tomentella-Thelephora*, Helotiales1, *Tricholoma* are the most dominant EM fungi lineages shared by two herbs, which is inconsistent with previous studies on various host plants and alpine ecosystems, of which *Cenococcum* is usually the most dominant EM fungal lineages in the roots of Polygonaceae plants (*Mühlmann, Bacher & Peintner, 2008*; *Gao & Yang, 2016*). We inferred that the difference may be due to the host specificity or the special fungal species bank in this study, which were caused by the special subalpine climate, but this is worthy of further study. However, the analysis of the correlation between the EM fungi lineages and environmental factors in this study showed that *Tomentella-Thelephora* was positively correlated with AK and MAP factors, *Inocybe* was negatively correlated with MAP, and *Pachyphloeus-Amylascus* was positively correlated with AN, Latitude, and TP and has a negative correlation with Longitude and TC. *Long et al. (2016)* research on EM fungi of *Pinus tabuliformis* in China shows that *Inocybe* is suitable for growing in arid areas. The possible reason is that the altitude is between 1,700–2,400 m, and the MAP is low, which affects the photosynthesis of the host plant and reduces the photosynthetic products distributed to the underground EM fungi, thereby, the diversity of EM fungi was affected to a certain extent.

PCoA ordination combined with PerMANOVA indicated that EM fungal communities were significantly different among the two herbs. Meanwhile, the Richness, Shannon and Chao1 index on the analysis of the alpha diversity among the two herbs show the same results. This is consistent with the previous research results of herbs (*Thoen et al., 2019*). For example, there are differences in the diversity and distribution of EM fungi detected in three high mountains in southwest China (*Gao & Yang, 2016*). There are significant differences in the community structure of EM fungi associated with *P. viviparum* between different nutrient habitats (*Mundra et al., 2015*).

According to the analysis of the above research results, it is found that the deterministic (environmental filtration) and stochastic (geographical distance) processes participate in the assembly process of herbs-associated EM fungal community in the subalpine area of central Inner Mongolia, and affected by ecological drift in the main stochastic process, which is inconsistent with previous studies on EM fungi (*Wang et al., 2021*).

One explanation may be that spatial distance hinders the spread of fungal propagules (*i.e.*, fungal fruit bodies, spores, and hyphae) (*Peay, Garbelotto & Bruns, 2010*; *Beck et al., 2015*). In addition, fungal species with different spread abilities may cause changes in the fungal community due to prior effects, which means that in terms of occupying habitat and using resources, fungi that arrive early have a stronger competitive advantage than those that arrive late (*Peay et al., 2012*). In this study, among the abiotic factors affecting the community structure of EM fungi, soil factors and climate factors were more important than geographical distance. This is inconsistent with previous studies. The analysis may be similar to the niche of two herbaceous plants in this study. No significant influence of spatial distance on community differences has been found. In previous studies, they have been proved to affect EM fungal communities, but this needs to be investigated in our

future studies. It should be noticed that different studies gave different results, this may be due to the various habitats, plant species and geographic scales. Thus, more studies involved specific habitats, plant species need explored in the future studies.

We should acknowledge that some analysis such as preference analysis or indicator analysis should be conducted to investigate biomarker fungi of each plant species, but site effect should be considered in these analyses. Additionally, as the two herb species harbored diverse EM fungi, we should use the culture-depend method to collect EM fungi, which can be used for ecological restoration. And we suggest the manager of the Chunkun Mountain (CKS) and Jiufeng Mountain (JFS) should reduce the grazing activities to protect the herbs, and thus maintain the EM fungal diversity in the semi-arid ecosystem.

## CONCLUSIONS

In summary, to our knowledge, this is the first time to investigate the diversity and community of EM fungal associated with *C. pediformis* and *P. divaricatum* in the subalpine of central Inner Mongolia, China. We found a high diversity of EM fungi in the two herbs, that is, 70 EM fungal OTUs that belonged to 21 lineages were reported in the present study. The most abundant lineage, such as *Tomentella-Thelephora, H*elotiales1, *Tricholoma*, show higher advantages in both herbaceous plants. Statistical analysis found that there are significant differences in the alpha diversity of EM fungi between different host plants. The EM fungal community contains a small number of dominant lineage and a large number of rare lineage. The PCoA ordination results show that there are significant differences in EM fungal communities between different herbs. Envfit analysis found that the environmental factor affecting community composition is OM and MAP. The results of NCM, $\beta$NTI and RC$_{bcary}$ indicate that the ecological drift in the stochastic process significantly affects the assembly of the EM fungal community. In this study, EM fungi were found *C. pediformis* and *P. divaricatum* in the subalpine meadows of CKS and JFS in central Inner Mongolia, and analyzed their fungal diversity, community composition and maintenance mechanism. This study can provide a certain reference for the research of herbaceous EM fungi and the special ecological relationship between herbs and mycorrhiza in alpine and subalpine environments, and provide a certain scientific basis for soil and water conservation, ecological environment restoration and sustainable development in subalpine areas of Inner Mongolia.

## ACKNOWLEDGEMENTS

We thank FaHu Li from the Jiufengshan (JFS) Inner Mongolia Daqingshan National Nature Reserve, and YiRan Yang from Chunkunshan (CKS) for help during sampling.

### Funding

This study was supported financially by the Natural Science Foundation of China (no. 32260006, 32260027), the science and technology project of Inner Mongolia Autonomous Region (no. 2019GG002), the Natural Science Foundation of Hebei Province (no.

C2020408015), the Inner Mongolia Natural Science Foundation (no. 2020MS03001), and the Science and Technology Project of Ordos (no. 2022YY008), and the Open Research Foundation of Yinshanbeilu Grassland Eco-hydrology National Observation and Research Station, China Institute of Water Resources and Hydropower Research (no. YSS2022012). The funders had no role in study design, data collection and analysis, decision to publish, or preparation of the manuscript.

### Grant Disclosures
The following grant information was disclosed by the authors:
Natural Science Foundation of China: 32260006, 32260027.
Inner Mongolia Autonomous Region: 2019GG002.
Natural Science Foundation of Hebei Province: C2020408015.
Inner Mongolia Natural Science Foundation: 2020MS03001.
Ordos: 2022YY008.
Yinshanbeilu Grassland Eco-hydrology National Observation and Research Station.
China Institute of Water Resources and Hydropower Research: YSS2022012.

### Competing Interests
The authors declare that they have no competing interests.

### Author Contributions
- Yongjun Fan conceived and designed the experiments, performed the experiments, analyzed the data, prepared figures and/or tables, authored or reviewed drafts of the article, provide experimental funds, and approved the final draft.
- Simin Xiang conceived and designed the experiments, performed the experiments, analyzed the data, prepared figures and/or tables, authored or reviewed drafts of the article, and approved the final draft.
- Jing Wang conceived and designed the experiments, prepared figures and/or tables, and approved the final draft.
- Xuan Zhang conceived and designed the experiments, performed the experiments, analyzed the data, prepared figures and/or tables, authored or reviewed drafts of the article, and approved the final draft.
- Zhimin Yu performed the experiments, authored or reviewed drafts of the article, and approved the final draft.
- Shupeng Zhu performed the experiments, authored or reviewed drafts of the article, and approved the final draft.
- Meng Lv performed the experiments, authored or reviewed drafts of the article, and approved the final draft.
- Lijun Bai performed the experiments, authored or reviewed drafts of the article, and approved the final draft.
- Luyu Han performed the experiments, authored or reviewed drafts of the article, and approved the final draft.

- Jianjun Ma conceived and designed the experiments, prepared figures and/or tables, and approved the final draft.
- Yonglong Wang conceived and designed the experiments, analyzed the data, prepared figures and/or tables, authored or reviewed drafts of the article, and approved the final draft.

## DNA Deposition
The following information was supplied regarding the deposition of DNA sequences:
The original sequences of 23 samples are available at the NCBI SRA database: SAMN26982404–SAMN26982426.

## Data Availability
The raw measurements are available in the Supplemental Files.

## Supplemental Information
Supplemental information for this article can be found online at http://dx.doi.org/10.7717/peerj.15626#supplemental-information.

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
