# Peer review of "First report of the ectomycorrhizal fungal community associated with two herbaceous plants in Inner Mongolia, China"

_PeerJ, doi:10.7717/peerj.15626_

## Round 0.1 · original submission · Major Revisions

Thank you for the submission to PeerJ journal on this interesting topic. We have received some critical comments demanding revision. Please update the title as suggested by reviewer #1. Please, present the details on community assembly of ectomycorrhizal fungi more clearly, and show the novelty of the research for the Inner Mongolia region.

Reviewer 1 ·

Basic reporting

No comment

Experimental design

No comment

Validity of the findings

No comment

Additional comments

This manuscript (Fungal diversity and community assembly of ectomycorrhizal fungi associated with two herbaceous plants in Inner Mongolia, China) is an interesting topic. Research questions are well defined and within the aims and the scope of the journal. The introduction is adequate and includes in suitable way the relevant publications. Materials are properly described. Methods are properly described and used in a way that is possible to replicate. The investigation is performed to good technical standards. A nicely conducted research with conclusions well supported by the results. However, the title of this manuscript is very similar to another article published by the same research group ( https://doi.org/10.3389/fmicb.2021.646821 - ´Fungal Diversity and Community Assembly of Ectomycorrhizal Fungi Associated With Five Pine Species in Inner Mongolia, China` - Front. Microbiol.). Thus, I recommend improving the title with regards to the novelty. By the way, I believe the authors might clarify better the hypothesis and the novelty of the study in the Introduction. Furthermore, I recommend increasing the length of the discussion by suggesting other analyses for studies in the future as well as citing recent references for the sentences in the introduction:

´Ectomycorrhizal (EM) fungi is crucial for growth and survival of a number of ecologically
and economically important plant species in boreal, temperate and arctic regions`;

´...and rely on the soil environment to obtain the nutrients needed for growth through host plants`;

´it is found that abiotic factors (soil factors, precipitation) are the decisive factors`

Reviewer 2 ·

Basic reporting

It’s better to polish the manuscript by a fluent English speaker thoroughly.
For example, “is” should be “are” in L123.
In addition, in “Supplementary Table 1. Sampling sites, plant species and environmrntal variables in present study”, “environmrntal” might be “environmental”.
Further, there might be no predicate verb in the first sentence of Analysis of Soil Properties session. (L146-148)

Could the author add some references to be a stronger manuscript, please? For example, it could be not difficult to add a reference to support a description that “Its plant division belongs to the North China broad-leaved 126 forest mountain area in the Eurasian grassland area” (L125-126). In addition, it’s common to add references to introduce some geography parameter (i.e., altitude) in an article (L126).

Both the article structure, and figures were professional. The raw data were shared well.

Could the author add their hypothesis in the last phrase of Introduction session?

Experimental design

The study lands on the aims and scopes of the journal.

It’s supposed that the author would like to study the effect of geographical distribution pattern on the diversity and distribution pattern of EM fungi and potential maintenance mechanism of EM fungi in herbaceous roots. However, it could be necessary to give a hypothesis in this study, making a more clear research question for readers.

Could the author give more details on root samples and soil samples collection, please?

In “Supplementary Table 1”, there were seven samples in JFS, and three samples in CKS. However, it said that there were four to seven individuals in each location. Could the author elaborate how many root samples and soil samples and replications in both JFS and CKS, please? It could be ambiguous in the present manuscript about sample collection. But it’s could be vital for a manuscript in Materials and Methods session.

Could the author give more details on Analysis of Soil Properties and add references for each analysis method, please?

Could the author find a similar study that used Kruskal-Wallis test as a reference, please? Because it is vital for a study to use a correct test method.

Validity of the findings

Some details could be necessary to confirm that the result was robust. Please see comments of 2 and 3 sessions.

Could the author analyze the correlation between the two EM fungal diversity of two herbs with environmental factors, please? This could give a new finding to discuss the effect of environmental factors on EM fungal in this study.

Could the author give a uncertainty in the end of the Discussion session, please?

Additional comments

If the author could add a sketches on study area and sampling, it would help readers understand the sampling work of this study and make a stronger manuscript.

Could the author give suggestion on future management for administers in the Discussion session, please?

“7.96±1.72 (mean±SE)“ could be “7.96 ± 1.72 (mean ± S.E.)” L297. The second ” (mean±SE)” can be deleted.

---

## Round 0.2 · Minor Revisions

Thank you for the manuscript update. It is better now, but yet some points have to be clarified. Please comment on sample size (to answer reviewer #2) - would it be enough for the study?

Reviewer 1 ·

Basic reporting

No comment.

Experimental design

No comment.

Validity of the findings

No comment.

Additional comments

I have read the revised version of this manuscript and the response letter. I noticed that the authors have incorporated my suggestions. Thus, I have no more comments and recommend acceptance.

Reviewer 2 ·

Basic reporting

The authors have responded to all comments. Unfortunately, there could be still one important point.

Experimental design

L154-155. The author argued that there were 23 root samples and 6 soil samples. However, there were 22 samples and 10 soil samples in supplemental Table 1. It’s vital to describe sampling correctly. Additionally, was there any replications for each sample? If not, there were only 2 soil samples for JFS (Polygonum divaricatum L.) and 1 soil sample for CKS (Carex pediformis var. pediformis), respectively. Obviously, this could not acceptable for PeerJ policy, because three replications for one sampling/sample are necessary for a field study. I believe the authors might clarify better the methods of both root and soil sampling in the Sample collection session. A sampling map (a sketch) could be useful to describe the sampling method and the study site that helps readers understand well.

Validity of the findings

No comments.

Additional comments

No comments.

---

## Round 0.3 · accepted · Accept

Thanks for the manuscript update and rebuttal letter. I see that last minor remark was considered. The manuscript is ready for publication now.